# Visual Smell: Learning Olfactory Representations for the Natural World

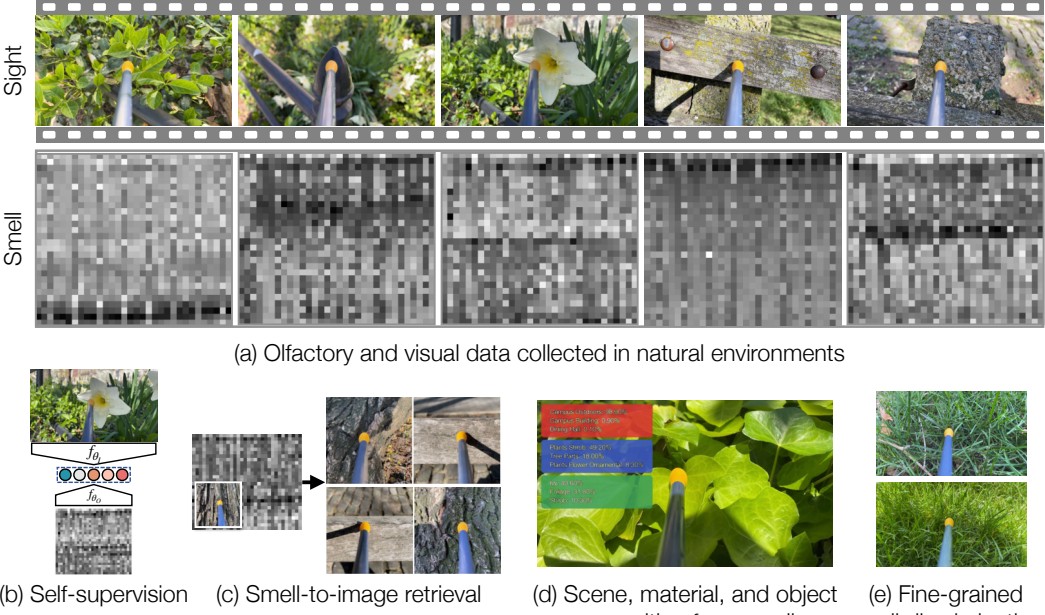

(a) Olfactory and visual data collected in natural environments

(b) Self-supervision (c) Smell-to-image retrieval (d) Scene, material, and object recognition from smell (e) Fine-grained smell discrimination

Figure 1: **Olfactory perception in the wild.** (a) We propose *Visual Smell*, a large and highly diverse dataset of natural olfactory signals and paired visual data. We show one sequence of images and smell signals that we obtained in a public park (one of dozens of scenes in our dataset). We then use this dataset for a variety of tasks: (b) learning cross-modal features between olfaction and images, (c) retrieving images based on their corresponding olfaction signals, (d) recognizing the scene, material, and object categories from an olfaction signal, (e) distinguishing different grass species.

## Abstract

Olfaction—the ability to sense volatile molecules in the air—is a key way that animals, and to a lesser extent humans, perceive the world. However, this rich "chemical world," is largely imperceptible to machines. One of the major obstacles to applying this approach to olfaction is the lack of suitable data and high quality feature representations. We address this problem in two ways. First, we propose a dataset of paired natural olfactory-visual data that is significantly more diverse and extensive than prior work. To capture it, we probe objects in natural indoor and outdoor environments with a smell sensor, while simultaneously recording video. Second, we use this dataset to learn self-supervised olfactory representations, by learning a joint embedding between visual and olfaction signals. We show that the resulting representation successfully transfers to a variety of downstream smell recognition tasks, such as recognizing different scenes, materials, and objects, and for making fine-grained distinctions between different types of grass.

# 1 INTRODUCTION

Olfaction—the ability to sense volatile molecules in the air—is a key way that animals, and to a lesser extent humans, perceive the world. Yet, this rich "chemical world", central to the sensory experience of many species, is largely imperceptible to machines.

This is in contrast to sight, sound, and touch, where recent advances in machine learning, particularly unsupervised and multimodal learning, have led to rapid improvements in machine abilities. One of the major obstacles to applying this approach to olfaction is the lack of suitable data. Existing datasets have largely been based on perceptual descriptors, rather than olfactory sensors, or are captured in lab settings. They are also not paired with other sensory modalities, such as vision, making it difficult to link olfaction to existing machine learning representations.

In this paper, we address this problem by capturing a large dataset of paired vision and smell. We visited dozens of indoor and outdoor scenes, such as parks, gyms, dining halls, libraries, and streets. In each one, we recorded naturally synchronized images and smells of their odorant objects *in situ*. The resulting dataset, which contains 7K samples from 3500 objects, is significantly larger and more diverse than other efforts (e.g., containing $70\times$ as many distinct objects as the lab-collected concurrent work of Feng et al. (2025a)).

We use the resulting dataset for self-supervised representation learning. Inspired by successful methods in other multimodal domains (Arandjelovic & Zisserman, 2017; Tian et al., 2020; Radford et al., 2021), we learn general-purpose olfactory features through contrastive learning, resulting in a joint embedding between visual and smell signals. Successfully learning this embedding requires matching smells to their corresponding sights in natural settings. This is a challenging task that is well-suited for pretraining. It requires bridging the natural variation between objects odors, as well as invariance to the the complex mixtures of smells and unconstrained airflow patterns that are present in real-world scenes.

We also use this dataset to establish a benchmark for in-the-wild smell perception, exploiting the paired visual data. We use this data to obtain pseudolabels for odorant objects, materials, and scenes, which we use to define corresponding recognition and smell retrieval tasks. We use these tasks to study the effectiveness of different olfactory representations. Our work suggests, first, that simple neural networks applied to the raw olfactory signal significantly outperform hand-crafted features considered in prior work. Second, we find that our learned feature representation provides useful pretraining for olfaction retrieval and recognition. To further evaluate the effectiveness of these features for fine-grained recognition, we propose a benchmark for distinguishing different species of grass.

We see this work as a step toward olfaction perception in the wild. While olfaction has traditionally been approached in constrained settings, such as quality assurance, there are numerous applications in natural settings. For example, as humans, we constantly use our sense of smell to assess the quality of food, identify hazards, and detect unseen objects. Moreover, many animals, such as dogs, bears, and mice, show superhuman olfaction capabilities, suggesting that human smell perception is far from the limit of machine abilities. Dogs, for example, may be able to track scent trails that are several days old and predict past and future events. They can distinguish visually identical percepts, such as identical twins (Hepper, 1988b), and breath samples from patients with disease or not (Buszewski et al., 2012) with high specificity and sensitivity.

Our contributions are as follows:

- We collect a dataset, *Visual Smell*, that is significantly larger, more diverse, multimodal, and naturalistic than existing olfaction datasets.
- We propose a benchmark for recognizing odorant objects, scenes, materials, and grass species from smell.
- We confirm that neural networks trained on raw olfactory signals outperform hand-crafted features.
- We show that joint embeddings between sight and smell produce high quality olfaction features.

## 2 RELATED WORKS

**Cross-Modal Supervision.** There have been a variety of different methods for supervising one sensory modality using another. Early work by De Sa (1993) proposed to use hearing to train vision through self-supervision. Ngiam et al. (2011) used a deep generative model to learn an audio-visual speech representation. In contrast to these works, we use our dataset for *olfactory* representation learning through cross-modal supervision with sight. Our work is closely related to audio-visual (Owens et al., 2016) and visual-tactile (Yuan et al., 2017; Yang et al., 2022; Dou et al., 2024) data collection efforts in which a human probes objects with a sensor while recording video. By contrast, we pair olfaction with multiple visual sensors. Recent work has learned a multimodal representation of taste (Bender et al., 2023). However, this approach is based on solely on text descriptions of wine, whereas we use a real signal from a sensor.

**Machine Learning for Olfaction.** Machine learning for olfactory sensing is an emerging field. Raw signals from electronic noses are high-dimensional and noisy, making data-driven methods attractive for uncovering structure. At the molecular level, psychophysical datasets enable models to predict perceptual attributes (Keller et al., 2017), and graph-based approaches propose a principal odor map (POM) (Lee et al., 2023a). Mixture studies are limited, showing approximate perceptual similarity (Snitz et al., 2013) and the existence of olfactory metamers (Ravia et al., 2020). Exploratory work has used mass spectra (Debnath & Nakamoto, 2020) and ion-mobility/e-nose data (Müller et al., 2019), but mostly under lab conditions. Recent work emphasizes the importance of calibrating olfactory neuroscience to natural concentration ranges (Wachowiak et al., 2025a), motivating the need for olfactory data in natural environments. In very recent concurrent work on arXiv, Feng et al. (2025a) collect a dataset of smells using an olfaction sensor. However, their approach is limited to a highly controlled lab setting and is relatively small scale (comprising 50 objects). By contrast, our work: 1) contains "in the wild" olfaction recordings in natural environments of smells, 2) contains paired multimodal signals, 3) is much more extensive. We also go beyond prior work by using our dataset for multimodal representation learning.

**Nonhuman Olfaction.** This work is inspired in part by the celebrated olfactory capacity of various nonhuman animals. Domestic dogs (*Canis familiaris*) in particular, are renowned for having an extraordinary sense of smell. Their ability is manifest in various detection tasks – identifying everything from the presence of bed bugs to landmines to owners' low blood sugar (Gadbois & Reeve, 2014) – but is also present in every untrained member of the species. Interestingly, diffusion tensor imaging of canine cortical regions reveals that olfaction is also structurally connected to vision: an extensive white matter pathway connects the olfactory bulb and the visual cortex Andrews et al. (2022). Anatomically, dogs are designed to be olfactory. They have hundreds of millions more olfactory receptor cells, the neural cells that begin the translation of VOCs into the perception of an odor, than humans do (Hepper & Wells, 2015). This enables them to detect more smells and more types of smells at lower concentrations. Their noses have separate routes for smelling and respiration, which enables airflow to arrive at the olfactory epithelium with every inhale (Craven et al., 2010). The olfactory bulb is two percent of their brain by volume – sixty times the relative size of the human olfactory bulb to the brain (Hepper, 1988a).

## 3 THE *Visual Smell* DATASET

We collect a large-scale dataset of natural olfactory-visual sensory data. Specifically, our dataset contains multimodal "smell-centric" data. Unlike previous efforts on smell in machine perception (Lee et al., 2023b; Feng et al., 2025b) and olfactory neuroscience Wachowiak et al. (2025b), which rely on controlled or synthetic environments and stimuli, our dataset is collected in the wild: humans (the authors) probe everyday objects in their natural environments using paired vision and olfaction sensors. This approach captures the range of naturally occurring odorant concentrations, a property that is key for modeling olfaction under natural conditions (Wachowiak et al., 2025b). We will publicly release the full dataset upon acceptance.

### 3.1 COLLECTING A VISUAL-OLFACTORY DATASET IN NATURAL SETTINGS

We now describe how we collected the dataset.

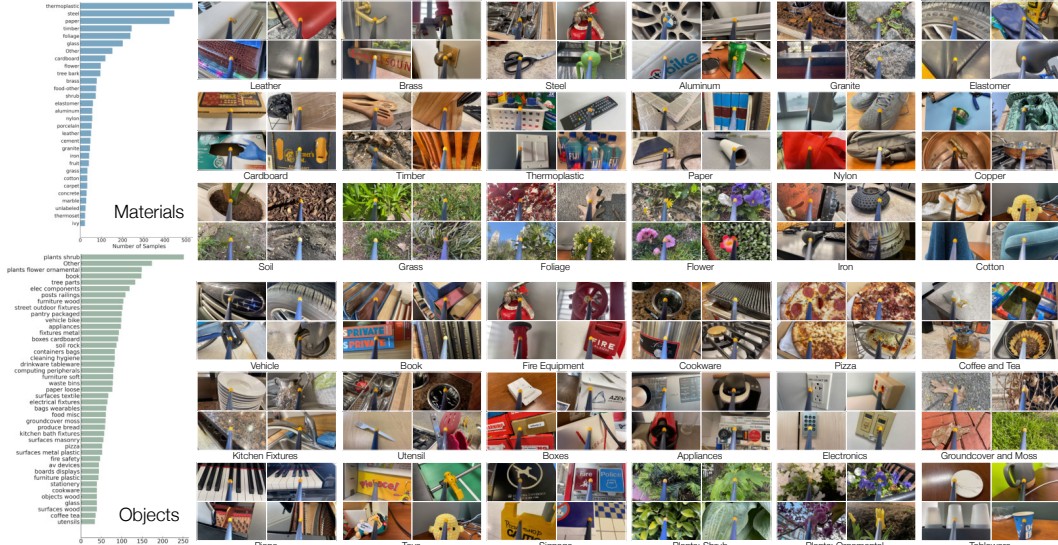

Figure 2: **The *Visual Smell* dataset.** We collect a diverse dataset of paired sight and olfaction. We visited many locations within a city and recorded a variety of scenes, materials, and objects. We show a selection of the captured images here, along with a distribution of pseudolabels, which are used for downstream evaluation. We demonstrate the utility this multimodal olfaction data by learning smell representations through cross-modal self-supervision. We provide an interactive preview of the dataset in the Appendix.

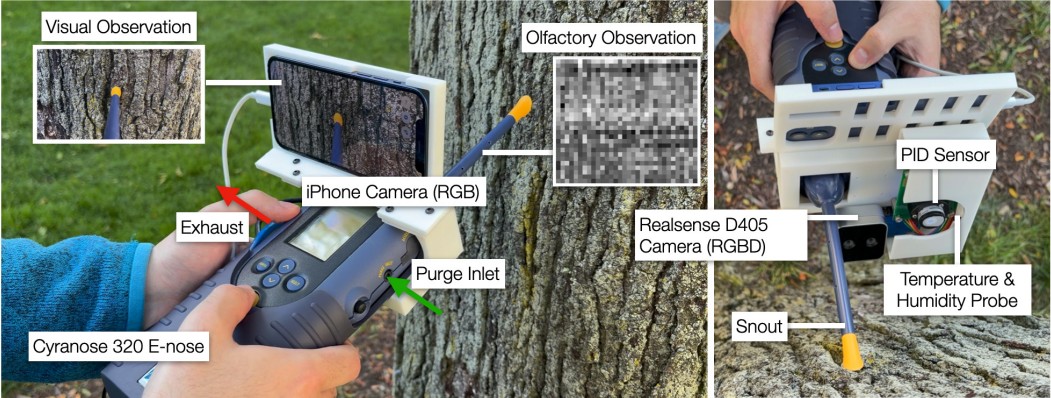

Figure 3: Olfactory-visual capture setup. Our setup is centered around the Cyranose 320 electronic nose, which outputs 32-dimensional signatures corresponding to different smell profiles. To capture paired RGB images of the olfactory source, we mount an iPhone 12 camera onto the Cyranose, angled toward the tip of the snout where olfactory measurements are taken. For additional complementary data, we integrate an Intel RealSense D405 camera for RGB and depth sensing, temperature and humidity probe, and a MiniPID2 PPM WR sensor for measuring ambient VOC concentrations.

**Hardware.** To collect olfactory and visual data in natural environments, we exploit the natural synchronization between smell and sight during an olfactory observation. As our olfactory sensor, we use the Cyranose 320 electronic nose, a popular sensor designed for hand-held use (Sensigent, 2000). Cyranose consists of a nanocomposite sensor array of 32 sensors. Each sensor responds to different chemical properties of volatile compounds that make up smell, without being specific to one volatile compound. We mount an iPhone 12 camera on Cyranose, angled to view the snout, where the olfactory measurement is collected. Cyranose operates at 2Hz, providing a 32-dimensional olfactory measurement at each timestep. Synchronized with Cyranose, the high-fidelity RGB camera captures the olfactory measurement at 1920×1080 resolution and 15 FPS. As complementary data for our dataset, we also record in synchronization a second RGB view and depth from an Intel Re-

alSense D405 at 15 FPS and 1280×720 resolution. Additionally, we record ambient temperature and humidity, and collect complementary ambient VOC concentrations olfactory measurements using a MiniPID2 PPM WR sensor. The PID olfactory measurements reflect the naturally occuring concentrations of smells by diffusion, rather than active sniffing. The additional RGB and depth observations provide a secondary view, from below the snout. The e-nose and all sensors are tethered real-time to a mobile station, consisting of battery, data storage, and compute to enable data collection across diverse settings, from parks, apartment settings, to streets. The complete capture set-up is shown in Figure 3.

**Capturing procedure.** An olfactory observation is the change in each of the 32 dimensional sensor dimensions relative to the ambient (baseline) olfactory observation from the environment. We must therefore first capture the baseline smell, followed by the smell of the object of interest. For olfactory measurement, Cyranose has two independent air pathways that can "sniff" outside air into its sensor chamber. The purge/baseline inlet, shown in Figure 3, on the side of Cyranose, pulls in ambient air, which leaves the sensor array through the exhaust outlet. Through this purge inlet-to-exhaust pathway, we first record the baseline smell for 10 seconds, receiving a $(14, 32)$ baseline matrix, representing each sensor for 14 timesteps. During this interval, air is drawn through the side port rather than the measurement inlet to avoid contamination from the target. Next, we record two samples through the main pathway: the snout. For data efficiency, we record two samples from different locations of each object, varying the olfactory and visual recording location. Both samples are 10 seconds, resulting in each $(14, 32)$ sample matrices. The *raw* olfactory data is thus a $(28, 32)$ matrix, which is the concatenation of the baseline and sample stages.

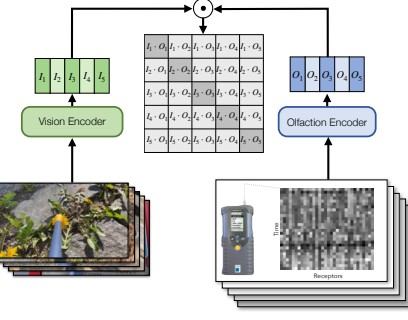

Figure 4: Multimodal representation learning through vision-olfaction alignment. Our model learns olfactory representations by correlating co-occurring visual and smell signals. A vision encoder processes RGB images while an olfaction encoder processes time-series sensor data from an electronic nose. The resulting embeddings are aligned via a cross-modal similarity matrix, encouraging correspondence between matching image-smell pairs.

**Existing olfactory representations.** One of our major goals is to learn useful representations for olfaction. Here, we review common practice. Previous work, including Cyranose device itself, have largely used a 32-dimensional vector called ***smellprint*** as the sensor response, which is computed from the raw smell matrix (Dutta et al., 2002). This vector is computed by applying Savitzky–Golay filtering (window length $w$, polynomial order $p$) independently to each sensor time series in both the baseline and sample stages. Let $R_{i,j}$ denote the (filtered) resistance of sensor $i \in \{1, \dots, 32\}$ at time index $j$. Let $B$ be the baseline indices, and $S = S_1 \cup S_2$ the union of the two sample windows. We compute per-sensor ambient level, sample peak, and smellprint as:

$$\text{Baseline (ambient) resistance:} \quad R_{0,i} = \frac{1}{|B|} \sum_{j \in B} R_{i,j}, \tag{1}$$

$$\text{Sample peak resistance:} \quad R_{\max,i} = \max_{j \in S} R_{i,j}, \tag{2}$$

$$\text{SmellPrint (relative response):} \quad \text{SmellPrint}_i = \frac{R_{\max,i} - R_{0,i}}{R_{0,i}}. \tag{3}$$

This 32-dimensional vector that summarizes the sensor response to odorants relative the ambient environment. Note that this representation discards large amounts of information, such as the 2nd-order statistics of the different sensors. Unlike prior work with Cyranose that has only used smellprint, we aim to use the raw olfactory signal for representation learning, using visual supervision. In Section 5, we compare olfactory-visual models trained with the raw signal to those trained with smellprint as input.

**Annotating the dataset.** To make it easier to visualize our dataset (see Figure 2), analyze our models, and train linear probes we used vision-language models automatically generate smell labels from paired visual data, effectively distilling knowledge from visual modality to olfactory networks.

This cross-modal supervision enables us to create vision pseudo-labels for fine-grained smell categories without requiring explicit human annotation. For scene recognition from smell, we assigned each data collection session into one of 8 scene categories. For material recognition, we used the Matador visual taxonomy of materials (Beveridge & Nayar (2025)). Using both views available in our dataset and this taxonomy as a closed set of categories, we generated material pseudo-labeled with VLMs. For object recognition, we manually wrote a closed set of 49 categories that spans our dataset, then generating vision pseudolabels with VLMs. Implementation details on the pseudo-label scheme are provided in the Appendix. Selected examples of material and object pseudo-labels are provided in Figure 2.

**Dataset analysis.** In Figure 2, we show the distribution of materials and objects, and provide qualitative examples from the dataset. We (the authors) collected across 60 data collection sessions over two months. For consistency in measuring smell, only two human data collectors performed the olfactory measurements. The dataset has 7K olfactory-vision pairs, 3.5K unlabeled objects, 70 hours of raw video from both cameras, and 196K timesteps of raw smell measurement (baseline and sample stage olfactory measurements), excluding the purge cycle. The dataset settings include several parks, university buildings, offices, streets, libraries, apartment settings, and dining halls - each location having multiple data collection sessions. Our dataset is relatively balanced between 41% outdoor and 59% indoor environments. In the Appendix, we provide an anonymous visualizer to the full dataset.

## 4 METHOD

### 4.1 MULTIMODAL SELF-SUPERVISED REPRESENTATION LEARNING

A key barrier in developing machine olfaction is the need for annotated training data. Humans have a relatively limited sense of smell compared to some animals, making it challenging to establish the label taxonomy and gather annotations on the scale required for effective machine learning. We instead propose learning rich olfactory representations from unlabeled examples, leveraging cross-modal associations between smell and sight.

Specifically, given our dataset of smell and corresponding visual data $\{\mathbf{x}_S^i, \mathbf{x}_I^i\}_{i=1}^N$, we learn olfactory and visual representations $f_{\theta_S}$ and $f_{\theta_I}$ by jointly training both encoders using a contrastive loss (van den Oord et al. (2019)):

$$\mathcal{L}_{\text{contrast}}^{I,S} = -\frac{1}{N} \sum_{i=1}^{N} \log \frac{\exp\left(f_{\theta_I}(\mathbf{x}_I^i) \cdot f_{\theta_S}(\mathbf{x}_S^i)/\tau\right)}{\sum_{j=1}^{N} \exp\left(f_{\theta_I}(\mathbf{x}_I^i) \cdot f_{\theta_S}(\mathbf{x}_S^j)/\tau\right)}, \tag{4}$$

where $\tau = 0.07$ is a small constant. Following Radford et al. (2021); Tian et al. (2020), we analogously define the smell to image loss $\mathcal{L}_{\text{contrast}}^{S,I}$, where the denominator sums over the visual modality. We minimize both losses:

$$\mathcal{L} = \tfrac{1}{2}\left(\mathcal{L}_{\text{contrast}}^{I,S} + \mathcal{L}_{\text{contrast}}^{S,I}\right). \tag{5}$$

By associating sight and smell (Figure 4), our goal is to learn a representation that can support the downstream interpretation of olfactory stimuli across multiple levels: low-level fine-grained differences, mid-level material properties, and high-level object and scene context.

### 4.2 DOWNSTREAM TASKS EVALUATION

To evaluate the quality of our learned olfactory representations, we perform a set of linear probing experiments. In this setting, we freeze the olfactory encoder trained with cross-modal supervision and train only a linear classifier on top of the learned embeddings. This setup allows us to directly measure the discriminative power of the representation without additional fine-tuning or task-specific optimization.

We probe across (A.4) scene, material, and object recognition from only smell and (5.3) fine-grained discrimination.

Concretely, given a frozen encoder $f_{\theta_S}$, we compute representations $\mathbf{z}_S = f_{\theta_S}(\mathbf{x}_S)$ for each smell sample $\mathbf{x}_S$. A linear classifier with parameters $\mathbf{W}, \mathbf{b}$ then predicts a label $\hat{y}$ via:

$$\hat{y} = \arg\max_c \ \text{softmax}(\mathbf{W}\mathbf{z}_S + \mathbf{b})_c, \tag{6}$$

where $c$ indexes respective categories in A.4 and 5.3. The linear layer is trained with a standard cross-entropy loss using the training split of our dataset, and accuracy is reported on the held-out test split. This evaluation isolates the contribution of the representation itself. Strong linear probe performance indicates that the representation encodes semantic information that is linearly separable, and thus useful for downstream tasks.

## 5 EXPERIMENTS

We evaluate our approach on a diverse set of benchmarks to assess the quality of olfactory representations learned through cross-modal supervision. Specifically, we study cross-modal retrieval between smell and vision, recognition tasks including scene, material, and object classification from smell alone, and fine-grained discrimination of olfactory signals. We report both quantitative metrics and qualitative results, highlighting the advantages of visual supervision for learning meaningful olfactory embeddings in natural settings.

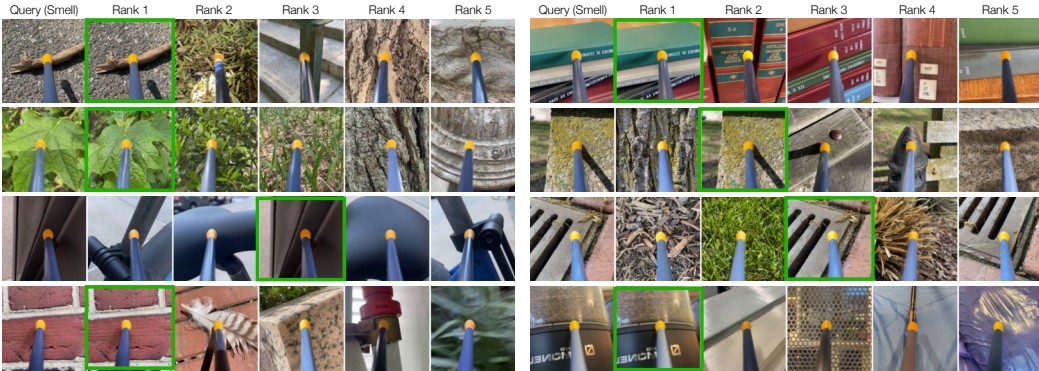

Figure 5: Qualitative results for cross-modal retrieval. Each row shows a reference smell query alongside the top-5 image retrievals predicted by our model. The ground-truth smell-image pair is highlighted in green.

### 5.1 CROSS-MODAL RETRIEVAL

**Setup.** For each query pair of smell and vision $\{\mathbf{x}_S^q, \mathbf{x}_I^q\}$ in our held-out test set, we sample a distractor set of images $D = \{\mathbf{x}_I^i\}_{i=1}^{N-1}$. We first embed the query pair into the shared olfactory and visual space to get $\{z_S^q, z_I^q\}$, where $z_S^q = f_{\theta_S}(\mathbf{x}_S^q)$ and $z_I^q = f_{\theta_I}(\mathbf{x}_I^q)$. We also embed every image $\mathbf{x}_I^i$ in $D$ into the same olfactory-visual space: $z_i = f_{\theta_I}(\mathbf{x}_I^i)$. We sort every image feature $z_i$ by its distance to the query smell feature $z_S^q$. If $z_I^q$ is closest to $z_S^q$, then it will have a rank of 1. Following Radford et al. (2021); Tian et al. (2020), we report Median Rank, Mean Rank, and Recall @ $K$, which measures the percentage of smell queries for which the matching image embedding is ranked in the top $K$ results.

**Results.** In Table 2, we compare CNN, MLP, and Transformer architectural variants of our olfactory encoder $f_{\theta_I}$ trained on the raw olfactory data to chance and $f_{\theta_I}$ that was trained on the lower dimensional summary metric smellprint. $f_{\theta_I}$ from the contrastive pretraining using smellprint performs better than chance in all metrics. However, training $f_{\theta_I}$ on the raw olfactory data leads to significant improvement compared to the smellprint encoder, independent of architecture. This showcases the richer information present in the raw olfactory data, unlocking stronger cross-modal associations between sight and smell. We further show qualitative results in Figure 5. Retrievals from our model often show semantic groupings. The odor of a book retrieves images of other books, the odor of leaves retrieves images of foliage. These examples suggest that the learned representation captures meaningful cross-modal structure. Retrievals also group by material properties.

Table 1: Recognizing Scenes, Objects, and Materials from Smell

| Method | Scene Recognition Acc. (%) | Material Recognition Acc. (%) | Object Recognition Acc. (%) |
|---|---|---|---|
| | 8 categories | 53 categories | 49 categories |
| Chance | 12.5 | 1.9 | 2.0 |
| **Visual Smell MLP + Linear Probe (smellprint)** | **32.46** | 1.96 | 4.96 |
| Rand. Weights MLP (smellprint) | 31.36 | 5.97 | 5.84 |
| **Visual-Smell MLP + Linear Probe (raw smell)** | **93.5** | **13.3** | **20.8** |
| Rand. Weights MLP (raw smell) | 91.9 | 8.16 | 16.5 |
| **Visual Smell Transformer (raw smell)** | **90.4** | **14.0** | **18.41** |
| Rand. Weights Transformer (raw smell) | 72.7 | 7.1 | 12.32 |
| **Visual Smell CNN + Linear Probe (raw smell)** | **94.5** | **12.3** | **19.8** |
| Rand. Weights CNN (raw smell) | 74.5 | 9.4 | 8.7 |

For instance, the odor of moss on a concrete bench retrieves images of moss on tree bark and on another bench, while the odor of a wooden stick retrieves images of groundcover and a tree bark.

## 5.2 RECOGNIZING SCENES, OBJECTS, AND MATERIALS FROM SMELL

**Setup.** Smell provides complementary information to other modalities, such as material properties and temporal object state. We therefore ask whether our olfactory representations encode structure at these levels. Using vision-derived pseudolabels for our dataset (Section 3), we evaluate the representations with linear probes as described in Section 4.2. For each task, we compare architectural variants of the smell encoder (MLP, Transformer, CNN) trained via olfactory–visual contrastive learning against the same encoders with random weights, as well as versions trained on smellprint features rather than raw sensory inputs.

**Results.** As shown in Table 1, each olfactory encoder trained with visual supervision outperforms the same architecture with random weights. Models trained on raw sensory inputs also achieve higher accuracy than the olfactory-visual model trained with smellprint inputs across scene, material, and object recognition. These results show that visual supervision substantially improves downstream recognition performance and that raw olfactory data provides richer signals than smell-prints for learning olfactory representations. In Figure 6, we showcase Top 3 predictions from linear probing our smell encoder, spanning diverse scenes, materials, and objects in our test set.

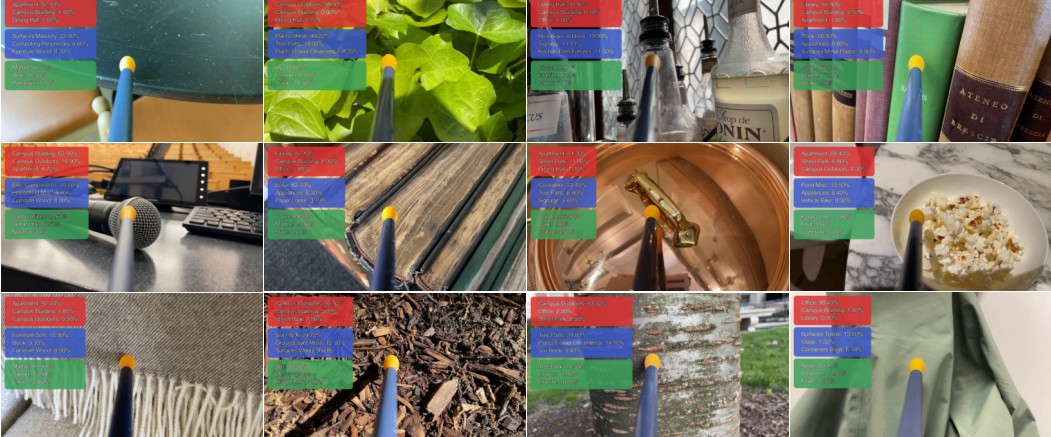

Figure 6: Examples of recognizing scenes, objects, and materials from smell, where the tabs show the Top 3 predictions from linear probing on the smell encoder. Predictions are organized by color: red indicates scene classification, blue indicates object classification and red indicates material classification.

## 5.3 FINE-GRAINED DISCRIMINATION

Dog's olfactory acuity (see section 2) enables trained dogs to excel in fine-grained olfactory discrimination. Even untrained dogs have been shown to distinguish their owner's t-shirt from the t-shirt of a stranger using a habituation-dishabituation paradigm Horowitz (2020). As a proof of principle,

Table 2: Cross-modal smell-to-image retrieval quantitative results ($N$=933).

| Smell Encoder | Mean Rank ↓ | Median Rank ↓ | Recall @ 5 (%) ↑ | Recall @ 10 (%) ↑ | Recall @ 20 (%) ↑ |
|---|---|---|---|---|---|
| Chance | 467 | 467 | 0.536 | 1.07 | 2.14 |
| MLP (Smellprint) | 375.9 | 329 | 2.04 | 3.43 | 6.22 |
| CNN (Raw Smell) | 118.4 | 41 | 12.9 | 21.1 | 32.6 |
| MLP (Raw Smell) | 159.5 | 56 | **17.3** | 24.2 | 33.8 |
| Transformer (Raw Smell) | **104.0** | **28** | 16.5 | **29.6** | **43.1** |

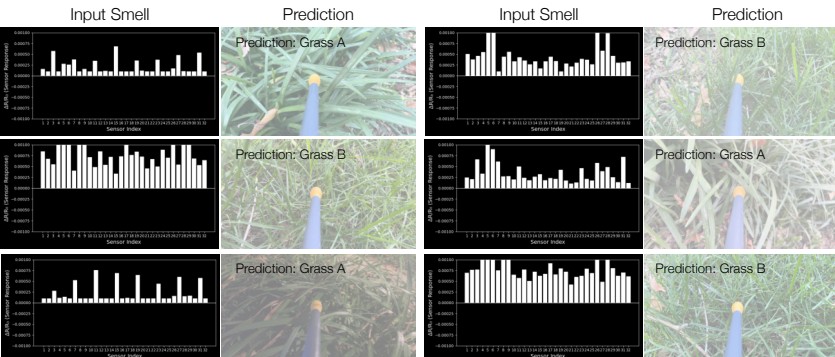

Figure 7: Examples of fine-grained discrimination. We showcase our model's ability to discriminate between two grass species meadow-grass (*Poa pratensis*) and Monkey grass (*Liriope muscari* by smell alone.

we first applied the same paradigm to 16 untrained dog subjects, who successfully discriminated between two visually similar grass species meadow-grass (*Poa pratensis*) and Monkey grass (*Liriope muscari*) by smell alone (see Appendix Figure 8).

**Setup.** We ask whether our learned olfactory representation can capture similarly fine-grained differences between the two grass species recorded at the same campus lawn, where they co-exist. To test this, we collected alternating samples of both species across six 30-minute sessions, yielding a balanced dataset of 256 examples. Following the procedure in Section 4.2, we trained a linear probe on the olfactory encoder learned through olfactory–visual contrastive learning and evaluated it on a held-out recording session of 42 samples.

**Results.** Table 3 reports binary classification accuracy for discriminating the two grass species. The olfactory encoder trained with smellprint input using olfactory–visual learning substantially outperforms chance and the same encoder with random initialization, underscoring the value of visual supervision. More importantly, training on raw olfactory data yields the highest accuracy—exceeding both random-weight baselines on raw

Table 3: Fine-grained discrimination from smell

| Method | Grass Acc (%) |
|---|---|
| Chance | 50.0 |
| Random Weights (smellprint) | 66.7 |
| Trained from Scratch (smellprint) | 85.7 |
| Ours: Linear Probe (smellprint) | 90 |
| Random Weights (Raw Smell) | 47.6 |
| Trained from Scratch (Raw Smell) | 52.4 |
| **Ours: Linear Probe (Raw Smell)** | **92.9** |

data and all variants trained with smellprint input. Even though our smell encoder trained with olfactory–visual pretraining was not trained end-to-end but only linear probed, it still outperforms training the smell encoder from scratch, both with smellprint and raw inputs. These results indicate that olfactory–visual learning with raw olfactory signals preserves more fine-grained information than learning with smellprints, and that visual supervision provides a crucial signal for exploiting this information. We showcase predictions from our model in Figure 7.

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
