# A APPENDIX

## A.1 PSEUDO-LABEL SCHEME

The following Python function is used to label objects using GPT-4o, where images are passed to GPT-4o along with a structured prompt to select the closest matching object category.

```python
def label_gpt_views(image_path1, image_path2, image_path3, image_path4,
    indexed_labels, labels):
    image_data1 = image_to_base64(image_path1)
    image_data2 = image_to_base64(image_path2)
    image_data3 = image_to_base64(image_path3)
    image_data4 = image_to_base64(image_path4)
    text_prompt = (
        "You are shown four images where a blue sensor probe with a
            yellow tip is pointing at the same object "
        "from different angles. This is the same target object being
            analyzed.\n\n"
        "Choose the best matching category label for this object from the
            list below.\n\n"
        "Respond with the NUMBER corresponding to the best label. Do not
            invent labels. "
        "If none are perfect, choose the closest match.\n\n"
        "If the image is gray (with white plus sign), choose unlabeled (
            number 0).\n\n"
        "Category options:\n" +
        "\n".join(indexed_labels) + "\n\n"
        "Respond with only the number. No text."
    )
    response = client.chat.completions.create(
        model="gpt-4o",
        messages=[
            {
                "role": "user",
                "content": [
                    {"type": "text", "text": text_prompt},
                    {"type": "image_url", "image_url": {
                        "url": f"data:image/png;base64,{image_data1}"
                    }},
                    {"type": "image_url", "image_url": {
                        "url": f"data:image/png;base64,{image_data2}"
                    }},
                    {"type": "image_url", "image_url": {
                        "url": f"data:image/png;base64,{image_data3}"
                    }},
                    {"type": "image_url", "image_url": {
                        "url": f"data:image/png;base64,{image_data4}"
                    }},
                ]
            }
        ],
        max_tokens=10,
        temperature=0
    )
    label_number = int(response.choices[0].message.content.strip())
    return label_number
```

Listing 1: Object labeling with GPT-4o.

The following Python function sends two image views of the same object to GPT-4o to determine the object's underlying physical material. The prompt includes detailed instructions and examples to avoid visual or semantic biases.

```python
def label_gpt_views(image_path1, image_path2, indexed_labels):
    image_data1 = image_to_base64(image_path1)
    image_data2 = image_to_base64(image_path2)
    text_prompt = (
        "You are shown two images where a blue sensor probe with a yellow
            tip is pointing at the same object "
        "from different angles. This is the same target object being
            analyzed.\n\n"
        "Your task is to identify the object's **main physical material**
            - not what it contains, what it's shaped like, or what it's
            used for.\n\n"
        "Choose the most appropriate material from the following list:\n"
        f"{', '.join(indexed_labels)}\n\n"
        "**Label what the whole object is actually made of.** Ignore
            gloss, color, texture, logos, or symbolic cues.\n\n"
        "**Examples:**\n"
        "- Any food item other than bread -> 'food-other'\n"
        "- A smooth paper cup -> 'paper'\n"
        "- A shiny white bowl -> 'porcelain' or 'thermoplastic' (depends
            on shape, stiffness, and context)\n"
        "- A juice dispenser labeled 'orange juice' -> 'thermoplastic',
            not 'fruit'\n"
        "- A brown padded chair seat -> 'leather', not 'terracotta'\n"
        "- A light-colored sidewalk slab -> 'concrete' or 'cement', not '
            asphalt'\n\n"
        "**Do not label based on:**\n"
        "- Color (e.g., orange != terracotta; gray != asphalt)\n"
        "- Function (e.g., juice bottle != fruit)\n"
        "- Gloss (e.g., shiny surface != glass)\n"
        "- Shape (e.g., cup shape != plastic)\n"
        "- Logos or printed text\n"
        "- Any object not being directly probed\n\n"
        "If you are absolutely certain it doesn't belong to any of the
            materials in the list, choose unlabeled. No text."
        "Now return **only the index** of the best matching material from
            the list as a number, without quotes. Never return any text
            such as 'I don't know' or 'unlabeled'."
    )
```

Listing 2: Material labeling from two image views using GPT-4o.

## A.2 DOG (*Canis familiaris*) FINE-GRAINED DISCRIMINATION BEHAVIORAL STUDY

As a proof of principle for our grass fine-grained discrimination experiment, we conducted a behavioral study with dogs using the habituation–dishabituation paradigm (Horowitz, 2020a). Sixteen subjects each completed four trials. Two grass species were placed in identical metal nosework canisters. During the first three trials, the same olfactory stimulus (Grass A) was presented until habituation occurred. After each trial, the sample was removed until the next presentation. Subjects' investigation time (sniffing) was behaviorally coded. In the fourth trial, a novel olfactory stimulus (Grass B) was introduced. A marked increase in investigation time on this trial indicated dishabituation, suggesting that dogs perceived the the novel stimuli (Grass B) as distinct from the previous one. Violin plots in Figure 8 show the distribution of sniffing durations, with each point representing a session and bold bars denoting mean and variability. These results demonstrate habituation to Grass A followed by dishabituation to Grass B, indicating robust discrimination ability between the two grass species.

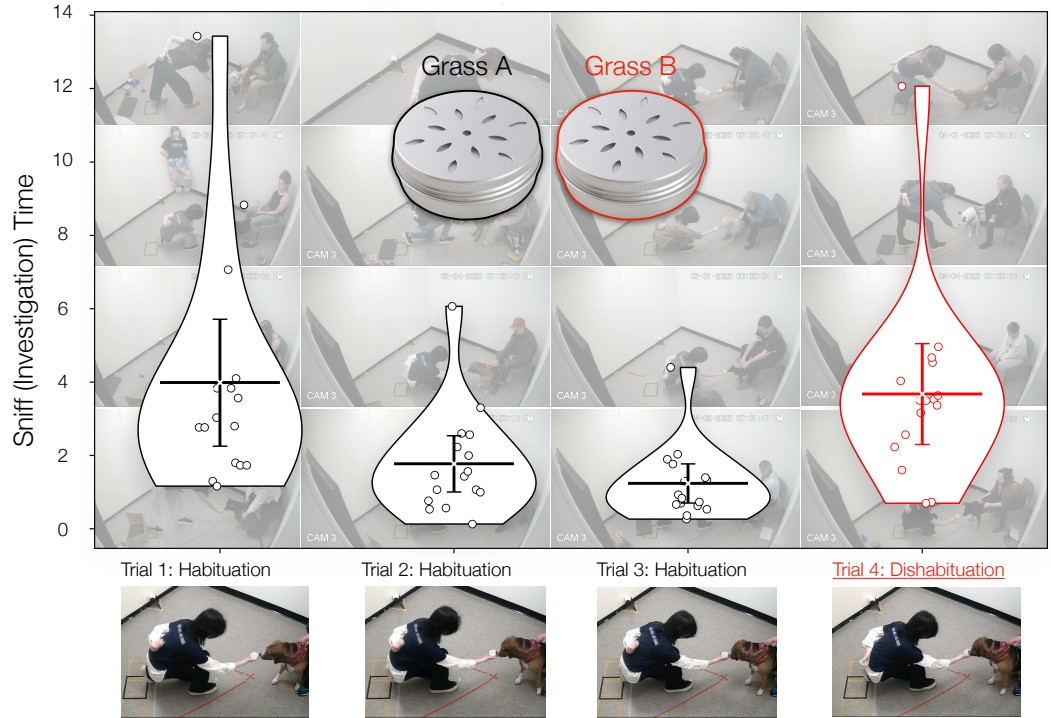

Figure 8: Dog olfactory discrimination experiment using habituation–dishabituation. 16 dogs are exposed to repeated presentations of Grass A (Trials 1–3), followed by a novel scent, Grass B (Trial 4). Increased sniffing time in Trial 4 indicates recognition of the new odor.

### A.3 VISUALIZING THE DATASET

To facilitate exploration of the dataset and support reproducibility, we provide interactive web-based previews of the collected olfactory-visual data. These visualizations allow reviewers to browse examples across different modalities and semantic labels, including object, material, and scene-level organization.

The dataset preview is hosted on an anonymous static mirror and includes the following interfaces:

**Main Overview.** A general summary of the dataset with samples and metadata across all data collection sessions: `https://demo.smellprojects.workers.dev/index.html`

**Objects View.** Visualization of object-centric samples with corresponding pseudo-labels and sensory traces: `https://demo.smellprojects.workers.dev/objects_new.html`

**Materials View.** Exploration of material-labeled samples derived from visual pseudo-labeling: `https://demo.smellprojects.workers.dev/materials.html`

These previews are intended for inspection purposes only, the full dataset and associated metadata will be released publicly upon publication.

### A.4 FINE-TUNING EXPERIMENT

We evaluate our olfactory-visual representation using linear probing, where the smell encoder is frozen. Here, we report the variants of the baselines where the smell encoder is trained from scratch.

Table 4: Recognizing Scenes, Objects, and Materials from Smell (training from scratch only)

| Smell Encoder | Scene Recognition Acc. (%) | Material Recognition Acc. (%) | Object Recognition Acc. (%) |
|---|---|---|---|
| | 8 categories | 53 categories | 49 categories |
| MLP full training (smellprint) | 42.2 | 3.84 | 3.33 |
| Transformer full training (raw smell) | 91 | 2.33 | 13.8 |
| CNN full training (raw smell) | 99.5 | 11.9 | 17.9 |

## A.5 APPENDIX: ADDITIONAL EXAMPLES OF RECOGNITION FROM SMELL

To complement the results in Section , we include additional randomly sampled examples in Figure 9 showcasing the ability of our smell encoder to recognize scenes, objects, and materials from olfactory input alone. These examples further demonstrate the generalization of our model across diverse contexts in the test set and highlight the semantic structure captured by olfactory representations trained with visual supervision. Each prediction is obtained via linear probing and is color-coded by task type: red for scenes, blue for objects, and green for materials.

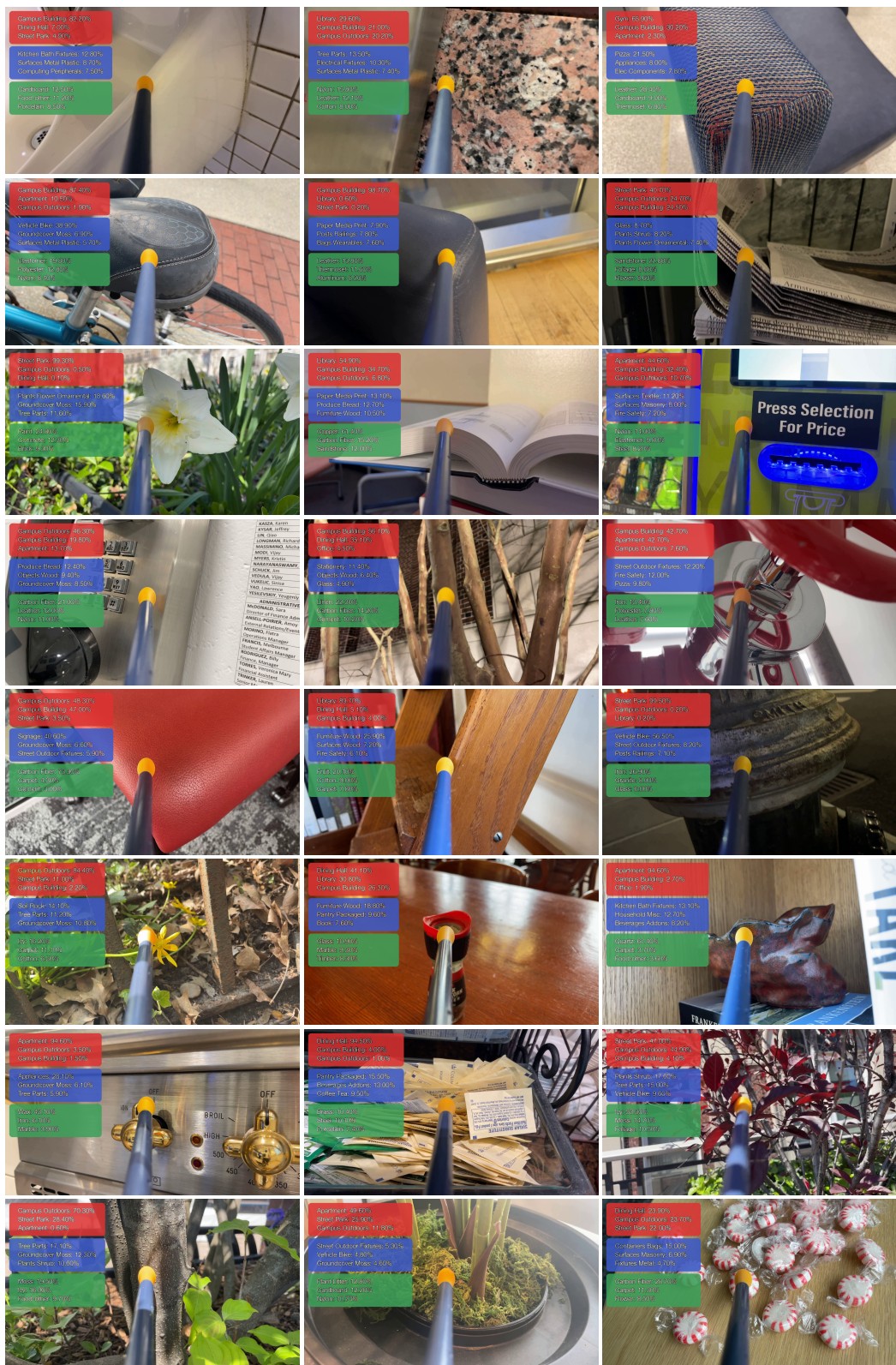

Figure 9: Non-cherry-picked examples of recognizing scenes, objects, and materials from smell. Each tab displays the top-3 predictions obtained via linear probing on the smell encoder. Prediction types are color-coded: red for scenes, blue for objects, and green for materials.