# OpenReview forum: "Visual Smell: Learning Olfactory Representations for the Natural World"
_ICLR.cc/2026/Conference — ICLR 2026 Conference Withdrawn Submission_

### Official Review · Reviewer_BgiV · 2025-10-27

**Soundness:** 3
**Presentation:** 3
**Contribution:** 3
**Rating:** 4
**Confidence:** 4

**Summary:**

In this paper, the authors contribute Visual Smell, a novel dataset containing 7K pairs of visual and chemical data pertaining to olfactory phenomena in-the-wild. The authors describe their experimental apparatus to collect the dataset and preprocess the data. Furthermore the authors employ their dataset to learn multimodal representations of visual and chemical data, using a standard self-supervised loss. Finally, the authors perform several downstream probing experiments to access the suitability of the learnt representations for cross-modal retrieval, object/scene recognition and fine-grained discrimination between similar classes.

**Strengths:**

- **Originality**: While large-scale datasets of olfactory phenomena exist and have been extensively used by the community to learn deep representations of olfactory data (e.g., [1]), the dataset presented here distinguishes itself in two ways: (i) it couples visual data to chemical information, which to the best of my knowledge is novel at this large scale; (ii) it is collected in-the-wild, and not in controlled environments. As such, I believe the work to be of sufficient novelty.

- **Quality**: I found the paper to be of high quality, with minimal typos and high-quality figures (one exception being Figure 6, which contains unreadable text). While not particularly novel, the use of self-supervision for learning multimodal representations is sound. The use of linear probes to evaluate the quality of the representations is also not particularly novel but, once again, a sound methodology. I also appreciated the discussion and evaluation throughout the paper on the different types of input representation for the chemical data.

- **Clarity**: In Section 3, the authors extensively describe their hardware setup to collect the dataset and preprocess the data. The description of the representation learning method and the experiments are also clear (despite some questions regarding the results, see below).

- **Significance**: As the authors point out, the lack of large-scale, high-quality olfactory data, collected in-the-wild, is currently a major bottleneck in olfactory machine learning. As such, this work is a part of ongoing data collection efforts by the community (see concurrent work, [2]). However, it distinguishes itself by the multimodal nature (chemical + image) of the data collected, thus it could become an interesting resource for the community.

**References**:
- [1] Lee, Brian K., et al. "A principal odor map unifies diverse tasks in olfactory perception." Science 381.6661 (2023): 999-1006.
- [2] Feng, Dewei, et al. "SMELLNET: A Large-scale Dataset for Real-world Smell Recognition." arXiv preprint arXiv:2506.00239 (2025).

**Weaknesses:**

- Electronic noses, such as the one employed by the authors, are well-known for having a limitations in regards to their use (see, for example, [1], [2]), such as degradation in performance due to humidity, temperature, sensor lifespan, etc... Despite this, there is no discussion on this paper regarding the limitations of the e-nose employed and the experimental conditions of the data collection. Can the authors elaborate on this?

- The constant presence of the e-nose snout in all image observations of the dataset might hinder the use of natural images in cross-modal tasks, for example for odor recognition from image data. Have the authors ran any experiment where they evaluated the use of natural images for this purpose (without the snout in the image)?

- Throughout the paper, the authors employ always image and chemical data in all downstream tasks. However, it would be important to evaluate the discriminative power of both modalities (in particular of the chemical one) for the proposed downstream tasks. What is the performance loss when considering each modality individually (for example, by learning representations in a self-supervised way, or using the downstream tasks supervision signal)?

- The results in Table 1 are quite unusual. Why do the "random weight" ablations have significantly higher than chance performance? I would assume that, for a large number of random initializations, the performance would tend to random chance.

- In Section 5.3., the authors describe an identification of different types of grass using untrained dogs. I suggest that the authors remove this experiment from the paper, as it contributes nothing to the message of the paper (there is no comparison between dog olfaction and machine olfaction, and it shouldn't be) and it raises questions regarding the treatment of the animals. This would allow the authors to have more space in the main document, for example, for a proper conclusion section.

**References**:
- [1]  Harper, W. James. "The strengths and weaknesses of the electronic nose." Headspace Analysis of Foods and Flavors: Theory and Practice (2001): 59-71.
- [2] Liu, Taoping, et al. "Review on algorithm design in electronic noses: Challenges, status, and trends." Intelligent Computing 2 (2023): 0012.

**Questions:**

See Weaknesses.

**Details Of Ethics Concerns:**

The authors present an experiment on dogs, yet there is no description of the conditions and treatment of these animals, neither in Appendix nor in the main text.

---

### Official Review · Reviewer_x3yB · 2025-10-27

**Soundness:** 2
**Presentation:** 1
**Contribution:** 2
**Rating:** 2
**Confidence:** 5

**Summary:**

Olfaction is notoriously hard for machine learning due to the neural processing of smells being highly nonlinear and the data being hard to collect. In this work, the authors work towards addressing this problem by proposing a new way to collect olfactory data; collecting a new dataset, and proposing a deep-learning architecture to model it. To this end, they combine a Cyranose 320 electronic nose with an iPhone camera and train a model jointly learn representations of the resulting smell profiles and images. They test their model on several classification tasks where they evaluate several design choices for their architecture.

**Strengths:**

- Novel data collection modality.

I find this part of the work truly exciting. As the Authors have mentioned, the available olfactory datasets are extremely limited (and, to add to that, barely compatible with earch other). This is because the olfactory data is both hard to generate and hard to perceptually quantify: Creating robustly reproducible smells is a hard engineering problem; describing them with robust reproducible vocabulary requires specialized training in prefumery. Unlike text or images, olfactory data is not produced as a byproduct of any other activity.

Thus the solution proposed here is neat. It allows to collect massive datasets of smells at a little effort: The collected smells will be ethologically relevant by design; they will be highly diverse; the video captioning will allow using vision-language and reasoning models to process the data.

- New dataset.

The Authors have collected a new dataset of smells using their proposed design. While currently, in size, it is similar to existing olfactory dataesets, typically featureng hundreds to thousands of data points, this one is extremely easy to scale because of the proposed design.

- Good literature overview.

I found the literature review clear and comprehensive, touching upong many types of the data and the models, available in the field.

**Weaknesses:**

It feels like the paper was written by two groups of people: While the dataset-related part is mostly nicely written, the machine-learning part seems to lack critical details.

- The details of the method are not sufficiently described

While the loss function is described and I could get some idea of what the model is through the Results Tables, I couldn't find a comprehensive description iof the model in the main text or in the supplement. Specifically, it looks like the models were trained using MLPs, CNNs, and Transformers on raw / processed outputs of Ciranose. While it overall makes sense for low-dimansional smell readouts, it's unclear if the same processsing was applied to the images / videos. At any rate, the details of the model and its training are necessary for the work's reproducibility.

- The details of the tasks are not sufficiently desribed

Again, through the Tables in the Results, I get an overall sense that the classification tasks encompasses 8 scenes, 53 materials, and 49 objects. It's less clear what the objects / materials are and how similar / different their smells may be or how many instances are present in each class. Some descriptions are provided as to how the background (scene) smells are cancelled but that raises questions about how the scences themselves are recognized.

- The relevance of the device to olfaction is not described.

While the device is called an Electronic Nose, it's unclear how relevant it is for olfaction. Recent datasets and models, cited by the Authors, by and large do not use this device. While I still like the approach to collecting the dataset, more discussion is needed to substantiate the use of this device or to define what we learn from its data and whether / how it is related to the human / animal olfaction, as suggested in the paper. As a counterexample, the air quality monitor that measures the concentrations of volatile particles of different size may also be sufficient to classify odor-emitting objects or materials but may have little to do witht the human olfaction.

- 3 out of 4 contributions needs futher clarifications or support

-- (2) the benchmark needs to be defined

-- (3) the claim is general and thus not supported. It is shown that a (not fully described) MLP performs better on the output of Ciranose compared to the Smellprint.

-- (4) the "quality of olfactory features" is not defined

**Questions:**

Plase add the missing details regarding the model, the task, the device used. Please make the list of the contributions concrete and show how your results support it. Until then, considering the overall lack of the details, even though the work is highly interesting, I sadly cannot recommend the acceptance of this work.

---

### Official Review · Reviewer_4Pzf · 2025-10-29

**Soundness:** 3
**Presentation:** 3
**Contribution:** 3
**Rating:** 4
**Confidence:** 4

**Summary:**

Paper presents a dataset of paired olfactory-visual data generated by probing objects in natural indoor and outdoor environments with a smell sensor, while simultaneously recording video. The dataset is used  to learn self-supervised olfactory representations, by learning a joint embedding between visual and olfaction signals.

**Strengths:**

The work pairs olfactory and visual signals, data is collected in the wild using an electronics nose. The idea is not original but the dataset is somewhat larger than what it currently exists. The paper is clearly written, method strait forward to understand. Relation to the state of the art is described in enough detail. In terms of significance, the topic is relevant and not fully explored yet. The definition of the model is not completely novel given it builds on existing work.

**Weaknesses:**

The major aspect is in using the electronic nose to generate olfactory data. Electronic nose is already calibrated to sense certain smells that takes away the nuances of natural smells. By targeting objects at certain distances when the data is generated eases the problem and it is not well described how distance from the source affects intensity, for example. The classification results are better and the ability to discriminate between two grass species is good, but the complexity of teh underlying problem is not well described.

**Questions:**

To what extent and how does the use of electronic nose affects the actual chemical perception of different olfactory signals?
Are there any limitations regarding the intensity of the source with respect to how far  from the source the sample was taken?
There is a significant amount of research in terms of sensing smell in the area of chemistry - how does the sensing part here relate to chemical research?

---

### Official Review · Reviewer_YctP · 2025-10-29

**Soundness:** 3
**Presentation:** 4
**Contribution:** 3
**Rating:** 6
**Confidence:** 4

**Summary:**

The authors present Visual Smell, a diverse dataset of visual-smell pairs gathered for objects using an olfactory sensor and an attached camera. Olfaction is a domain that is often bottlenecked by data availability, and the authors’ contribution is primarily in the release of this dataset. To evaluate their dataset, the authors adopt a contrastive learning approach on the visual and olfactory representations, and show its effectiveness for object, scene and material recognition, especially when the model is trained on raw olfactory data gathered by the sensor rather than human engineered features. While the architectures pursued are relatively simple, the authors effectively demonstrate their point that models trained jointly on visual and smell elevate model performance. I am leaning towards accepting this work for publication in ICLR, but the authors should address some critical concerns that I will mention below that will help lend further proof to the claims made within the work.

**Strengths:**

The authors essentially conduct fieldwork and gather the visual-smell pairs across a wide variety of objects, materials and scenes. This is no small task, and the dataset appears to be diverse enough to be of broad appeal. The authors also highlight another case of the bitter lesson and beat prior state-of-the-art human-engineered features by using deep networks on raw olfactory signals to learn better representations. Beyond providing the dataset, the authors also showcase some ways in which the dataset can be utilized, and provide some benchmark values for the performance of deep networks trained using a contrastive learning approach.

**Weaknesses:**

I feel like the benchmarks could have been stronger. The authors show that the model significantly outperforms human engineered features, but there’s no clear “gold standard” for what is considered accurate enough. In Table 1, while the scenes appear relatively diverse and non-overlapping, I wonder if top-1-accuracy is too strict for both material and object recognition.

The authors also do not state how they split the data into training and test sets -- the performance of the model can be strongly overestimated if the test set is significantly within the training set’s distribution. It would be important to show some level of generalizability for their trained models to further highlight the utility of models trained using these approaches.

I would also have liked to see a critical evaluation of the Cyranose platform that the authors use in this dataset, and its limitations or caveats.

On a final note, it would have been interesting to see some sort of human evaluation in visual-smell matching, though the olfactory mixture + scene is probably far too complex to reproduce in a controlled setting.

**Questions:**

1) How is the temporal component of the receptors accounted for in non-sequence based models? Is it averaged over time? Does it make sense that the transformer performs worse in scene and object recognition despite its sequential nature?
2) While I am convinced that this seems like proper protocol for measuring the olfactory observation, how valid is the assumption that it is correct to remove highly volatile components that might have belonged to the object of interest and treat it as part of the “ambient air”?
3) Along the same line, how much variance in the olfactory signals exists -- in terms of the olfactory properties of similar objects but in different environments?
4) “Other” seems to be a significant proportion of your materials and objects labelled by the VLM in Figure 2 -- how is this handled?

---

### Note · Authors · 2025-11-13

**Comment:**

We thank the reviewers for their time and constructive comments. We will revise the paper based off the comments and submit to another venue.

**Withdrawal Confirmation:**

I have read and agree with the venue's withdrawal policy on behalf of myself and my co-authors.